# Sensor Clustering Using a *K*-Means Algorithm in Combination with Optimized Unmanned Aerial Vehicle Trajectory in Wireless Sensor Networks

**DOI:** 10.3390/s23042345

**Published:** 2023-02-20

**Authors:** Thanh-Nam Tran, Thanh-Long Nguyen, Vinh Truong Hoang, Miroslav Voznak

**Affiliations:** 1Data Science Laboratory, Faculty of Information Technology, Ton Duc Thang University, Ho Chi Minh City 700000, Vietnam; 2Faculty of Information Technology, Ho Chi Minh City University of Food Industry, Ho Chi Minh City 700000, Vietnam; 3Faculty of Computer Science, Ho Chi Minh City Open University, Ho Chi Minh City 700000, Vietnam; 4Faculty of Electrical Engineering and Computer Science, VSB-Technical University of Ostrava, 17. listopadu 2172/15, 708 00 Ostrava, Czech Republic

**Keywords:** wireless sensor network (WSN), unnamed aerial vehicle (UAV), optimal UAV positioning, *K*-means clustering, gap statistic method, centroid-to-next-nearest-centroid (CNNC) trajectory

## Abstract

We examine a general wireless sensor network (WSN) model which incorporates a large number of sensors distributed over a large and complex geographical area. The study proposes solutions for a flexible deployment, low cost and high reliability in a wireless sensor network. To achieve these aims, we propose the application of an unmanned aerial vehicle (UAV) as a flying relay to receive and forward signals that employ nonorthogonal multiple access (NOMA) for a high spectral sharing efficiency. To obtain an optimal number of subclusters and optimal UAV positioning, we apply a sensor clustering method based on *K*-means unsupervised machine learning in combination with the gap statistic method. The study proposes an algorithm to optimize the trajectory of the UAV, i.e., the centroid-to-next-nearest-centroid (CNNC) path. Because a subcluster containing multiple sensors produces cochannel interference which affects the signal decoding performance at the UAV, we propose a diagonal matrix as a phase-shift framework at the UAV to separate and decode the messages received from the sensors. The study examines the outage probability performance of an individual WSN and provides results based on Monte Carlo simulations and analyses. The investigated results verified the benefits of the *K*-means algorithm in deploying the WSN.

## 1. Introduction

Deployments of wireless sensor networks (WSNs) are increasing because of their beneficial applications. For example, WSNs can be deployed to monitor or collect environmental data (meteorological information such as precipitation, wind speed and direction, air pressure, humidity, temperature, etc.) in remote or difficult terrain [1,2,3,4,5,6]. A major challenge in deploying a WSN is distributing a large number of wireless sensors over a large and complex geographical area. Wireless sensors are generally low cost, have low power consumption and are highly flexible in their application. However, transmitting a signal from a wireless sensor directly to a control centre presents an important challenge [7], especially if a large number of wireless sensors are deployed to directly collect data. Using terrestrial infrastructure for the purpose of collecting data from wireless sensors is impractical because of high deployment costs and a low flexibility. A potential solution to this problem is using an environmental monitoring system which dispatches an unmanned aerial vehicle (UAV) to a geographical area to retrieve the collected sensor data.

To deploy a WSN, Heinzelman et al. [8] proposed a low-energy adaptive clustering hierarchy (LEACH), a clustering method now considered the most well-known clustering protocol for WSNs. In a hierarchical topology, clusters contain two types of node: cluster members and cluster heads. Member nodes are grouped into different clusters, and in each cluster, a single node is designated a cluster head. The cluster head has the most important role in the cluster, tasked with receiving signals from cluster members and forwarding those signals to other cluster heads [9] or the base station [10].

UAVs have gained increasing consideration as aerial relays which deliver mobility and on-demand wireless connections in areas with complex topography and no network coverage. A UAV’s online time, however, is limited by its own on-board energy limitations. The evolution of UAV-assisted WSNs is compelling the scientific community to search for new ways of performing energy harvesting (EH) from external power sources to prolong the online time of UAVs. A variety of effective solutions have been proposed, grouped according to two main types of technique, namely, simultaneous wireless information and power transfer (SWIPT) and techniques for determining the optimal positions for the UAV.

Radio frequency EH shows promise as a potential solution for UAV-assisted WSNs. Initial studies on radio frequency EH were used in a technology termed wireless power transfer (WPT) to recharge the wireless sensors in the WSN. A new radiofrequency EH technique, termed SWIPT, introduced significant benefits to WPT [11]. Many authors have studied SWIPT over the last two decades [12,13,14,15], investigating the performance difference between time switching and power splitting in SWIPT protocols [16]. The current study applied a time-switching protocol because it supports a phase for EH.

Applied to all beyond 5G/6G wireless communications, nonorthogonal multiple access (NOMA) provides massive connections, low latency and high reliability [14,17,18,19,20,21]. In the current study, we took advantage of NOMA’s benefits, applying NOMA at the UAV to superimpose coding of the data received from wireless sensors and to forward this superimposed signal to a mobile data centre.

### 1.1. Motivation

The use of machine learning in practical applications is escalating. The authors in [6] applied an artificial neural network (ANN) for sensor clustering. By contrast, some wireless sensors in the study in [6] were clustered as separate single-member clusters. In [22], the authors proposed a distance- and energy-constrained *K*-means clustering scheme (DEKCS) for cluster head selection to prolong the lifetime of underwater WSNs. With this new clustering algorithm, a prospective cluster head was selected according to its position in the cluster and its residual battery level. The authors dynamically updated residual energy thresholds set for prospective cluster heads to ensure that the network fully depleted its energy before disconnection. In this manner, cluster heads could be drained of energy and become inactive/dead sensors. The current study applied the *K*-means algorithm and the gap statistic method first introduced in [23] to obtain an optimal number of subclusters. To our best knowledge, the gap statistic method has not been applied for WSN clustering in any previous study.

In [24], the authors examined a UAV-assisted data collection WSN. The UAV’s trajectory was optimized by applying the travelling salesman problem. Note that in [1], the UAV in the proposed network visited every wireless sensor, while in [24], the optimal serving order for sensors was determined according to a standard travelling salesman problem algorithm, which can be optimally solved with the efficient cutting-plane method (i.e., the shortest path from the start point to the end point). The authors also proposed an algorithm which used the pattern search method to solve the problem of optimizing the UAV position and sensor uploading power. In [1], the UAV could be exhausted as a consequence of long flight distances. The authors in [24] used a UAV that navigated the shortest path from the start point to end point, but it consequently ignored/missed some wireless sensors. In another study, the authors addressed the UAV’s trajectory problem by jointly optimizing the UAV’s velocity, hovering positions and visiting sequence [25]. The scientific community is very interested in studying UAVs’ trajectories for the significant potential gains in aerial network performance. Researchers have applied several types of trajectory, for example, straight trajectory [26,27], circular trajectory [10,28,29] and spiral trajectory [30,31,32]. The authors in [25] introduced an interesting UAV trajectory scheme (Figure 4), where the UAV visited all *N* monitoring areas and then found suitable positions to transmit the collected data. In [25], the UAV collected data in four stages: (i) UAV data collection flight, (ii) UAV data collection processing, (iii) UAV data transmission flight and (iv) UAV data transmission processing. That UAV’s operating schedule is illustrated in Figure 5 [25].

The current study proposes the use of a UAV for its high mobility, quick implementation and low cost. The main drawback to a small-sized, lightweight aircraft such as a UAV, however, is its limited on-board energy. UAVs are therefore not suitable for flying close to each sensor to collect data, as proposed in [25]. The current study therefore investigated the application of a *K*-means algorithm to cluster wireless sensors into multiple, optimized subclusters.

### 1.2. Contribution

Inspired by the studies mentioned in the previous section, we employed a UAV as an aerial relay to provide a sustainable, functional solution for a WSN. The main contributions of the current study are:The use of three-dimensional Cartesian coordinates for a WSN which contains a random number of randomly distributed wireless sensors.The decomposition of the UAV trajectory optimization into two subproblems: (i) the global WSN cluster is divided into multiple subclusters whose number is optimized with unsupervised machine learning which applies *K*-means clustering in combination with the gap statistic method; (ii) a centroid-to-next-nearest-centroid algorithm is then applied to find the shortest path for travel through every subcluster.An analysis of the system performance of the WSN over Rayleigh distributions and a presentation of the derived closed-form expressions for the outage probability at the UAV and mobile base station.Outage probability results for the UAV and mobile base station derived from Monte Carlo simulations and verified with an analysis.

The remainder of the paper is organized as follows: Section 2 introduces the WSN model, wireless sensor clustering algorithm, joint UAV trajectory, free-space channel modelling, and joint UAV operating schedule; Section 3 provides an analysis of the WSN’s performance based on outage probability and presents the closed form expressions for outage probability at the UAV and mobile base station; Section 4 examines and plots the investigated results; Section 5 discusses conclusions.

For clarity, Table 1 presents the notation used in the paper.

## 2. WSN Model

The current study examines a general WSN with a randomly distributed number of wireless sensors. Figure 1 depicts a WSN with a random number of sensors N=42 positioned at the Cartesian coordinate $\left(x,y,z\right)$ in three dimensions. Let us assume that a mobile base station *B* is positioned at $B\left(0,0,0\right)$ and each wireless sensor $S_{n}$ for $n=\left\{1,\ldots ,N\right\}$ is positioned randomly at coordinate $S_{n}\left(x,y,0\right)$, where $x=\left\{0.1,\ldots ,1\right\}$ and $y=\left\{0.1,\ldots ,1\right\}$ as shown in Table 2. For simplicity, we assume that the wireless sensors and mobile base station are positioned relative to a flat earth.

**Definition** **1.**
*We denote the global set C as containing all wireless sensors. $\left|C\right|$ returns N, the total number of wireless sensor nodes (i.e., $\left|C\right|=N$). Let us assume that the data observations (i.e., wireless sensor positioning) are clustered into K subclusters, i.e., $C⊇C_{1}\cup \ldots \cup C_{K}$ and $N=\left|C\right|=\left|C_{1}\right|+\ldots +\left|C_{K}\right|=\sum_{k=1}^{K}N_{k}$, where $N_{k}=\left|C_{k}\right|$.*


Figure 1 illustrates a random distribution of wireless sensors. Each wireless sensor is allocated a given index by the subscript *n*, where $n=\left\{1,\ldots ,N\right\}$ and a lower index *n* has a higher priority. For clarity, sensor $S_{n}$ has a higher priority than sensor $S_{n+1}$ (e.g., sensor $S_{1}$ has a higher priority than sensor $S_{2}$).

### 2.1. WSN Clustering

**Remark** **1.**
*Because the optimization problem is complex, we propose breaking it down into several subproblems and observing the random distribution of wireless sensors over a large geographical area, as illustrated in Figure 1. The global wireless sensor cluster can be divided into multiple subclusters. The number of subclusters can be optimized by applying the gap statistic method, and the wireless sensors can be assigned to a subcluster using the K-means algorithm. To solve these problems, we propose a solution in Proposition 1.*


**Proposition** **1.**
*The optimal number of subclusters is yielded as follows:*


*Observing the latitudes (x-axis) and longitudes (y-axis) of the wireless sensors, we determine the optimal number of subclusters $K\leftarrow k_{optimal}$. The gap statistic method is applied to the number of subclusters k to compute the corresponding total within the intracluster variation $W_{k}$, i.e., the sum of squares function, given by*

(1)
$$\begin{array}{c} W_{k}=\sum_{\kappa =1}^{k}\frac{1}{2\left|C_{\kappa }\right|}\sum_{S_{i},S_{j}^{}\in C_{\kappa }}d_{S_{i},S_{i}^{}}, \end{array}$$

*where $\kappa =\left\{1,\ldots ,k\right\}$, $\left|C_{\kappa }\right|$ returns the number of wireless sensor nodes in cluster $C_{\kappa }$ and $d_{S_{i},S_{j}^{}}$ is the squared Euclidean distance of all pairwise sensor nodes in the cluster $C_{\kappa }$ for $S_{i},S_{j}\in C_{\kappa }$ and i≠j. It is important to note that we assume $k_{min}=4$ and $k_{max}=\frac{\left|C\right|}{k_{min}}=\frac{N}{k_{min}}$, where $\left|C\right|$ is the number of observations (or the number of sensors within the global cluster C). Let us briefly consider factors $k_{min}$ and $k_{max}$. We define $k_{min}=4$ to prevent a uniform distribution of wireless sensors positions throughout the area; the gap statistic method thus returns the optimal number of clusters $k_{optimal}=1$. For example, Figure 2a,b in [23] plots the distribution of sensors spread throughout a region and the corresponding optimal number of clusters at K=1, respectively. However, we also define $k_{max}=\frac{N}{k_{min}}$ to prevent each wireless sensor owning a private cluster, a problem that would lead to a UAV visiting every wireless sensor to collect data.*

*Reference data sets *Ω* with a random uniform distribution are generated. Each reference data set *ω* of these reference data sets *Ω* is clustered with a variable number of clusters $k=\left\{k_{min},\ldots ,k_{max}\right\}$. The corresponding total is computed within the intracluster variation $W_{\kappa \omega }^{}$ given in the dispersion metrics for $\kappa =\left\{1,\ldots ,k\right\}$ and $\omega =\left\{1,\ldots ,\Omega \right\}$.*

*The estimated gap statistic is computed as the deviation of the observed $W_{k}$ value from its expected value $W_{\kappa \omega }$ under the null hypothesis $Gap\left(k\right)=\frac{1}{\Omega }\sum_{\omega =1}^{\Omega }\log\left(W_{\kappa \omega }^{}\right)-\log\left(W_{k}\right)$. Let $l=\frac{1}{\Omega }\sum_{\omega =1}^{\Omega }\log\left(W_{\kappa \omega }^{}\right)$. The standard deviation (sd) of the statistics is then computed, given by $sd_{k}=\sqrt{\frac{1}{\Omega }\sum_{\omega =1}^{\Omega }{\left(\log\left(W_{\kappa \omega }^{}\right)-l\right)}^{2}}$.*

*Using the gap statistic method, the smallest value of κ is selected as the optimal number of clusters, the gap statistic being within one standard deviation of the gap statistic at κ+1, given $k_{optimal}=\min\left\{k\right\}$ and $Gap\left(k\right)\geq Gap\left(k+1\right)-\theta_{k+1}$, where $\theta_{k+1}=sd_{k+1}\sqrt{1+\frac{1}{\Omega }}$.*



For example, Figure 2 indicates the optimal number of clusters at $k_{optimal}=4$, determined by the gap statistic algorithm according to the randomly positioned sensor nodes shown in Figure 1.

The *K*-means algorithm was used to calculate the position for each centroid, with an optimal number of clusters $K\leftarrow k_{optimal}$. The computed centroids of four subclusters (K=4) are listed in Table 3. Figure 3 illustrates all wireless sensor nodes after clustering to *K* subclusters. After clustering, each sensor node is grouped into a subcluster; for example, $S_{1}^{3}$ indicates that sensor $S_{1}$ is a member of subcluster $C_{3}$ (Figure 3).

### 2.2. Joint UAV Trajectory

**Definition** **2.**
*We introduce a novel joint UAV trajectory algorithm to compute the centroid to next nearest centroid.*


**Problem** **1.**
*Operating as a flying relay, the UAV has the advantage of a high mobility and is able to fly close to wireless sensors to receive and forward a superimposed signal. This, however, leads to a long flight path, and the other wireless sensors must wait to be served. We minimized the flight time/path of the UAV according to the cluster centroid positions shown in Table 3. How to obtain the shortest flight path is outlined in Proposition 2.*


**Proposition** **2.**
*The centroid-to-next-nearest-centroid trajectory was computed as follows:*


*Step 1: To determine the nearest centroid from the mobile base station B, we calculate the smallest pairwise Cartesian distance from the mobile base station to each subcluster centroid.*

*Step 2: The UAV selects the next nearest cluster centroid. In this case, the UAV considers candidate centroids without regard to any of the previously selected cluster centroids in $\tilde{C}$. It is important that the centroids contained in the visited set $\tilde{C}$ be removed from the candidate list to prevent the UAV returning to the previous subcluster $\tilde{C}$. The UAV repeats Step 2 (i.e., $C\setminus \tilde{C}\neq \emptyset$) until the list of candidate subclusters is empty (i.e., $C\setminus \tilde{C}=\emptyset$).*



Without loss of generality, we examined a single round trip of the UAV. Table 4 lists the next nearest subcluster centroids determined from the above selection strategy. The results in Figure 3 indicate that subcluster $C_{1}$ was the nearest to the mobile base station *B* compared to the other subclusters. Subcluster $C_{1}$ was therefore selected at block period time T=1. The UAV visited subcluster $C_{1}$ first to collect data from all sensor members in subcluster $C_{1}$. The visited set $\tilde{C}\leftarrow \tilde{C}\cup C_{1}$ was then updated. After all data from the sensor members in subcluster $C_{1}$ were collected, the UAV selected subcluster $C_{3}$ because it contained the next nearest subcluster centroid. The UAV then visited subcluster $C_{3}$ at global time period T=2 to collect data from all sensor members in the subcluster. The visited set $\tilde{C}\leftarrow \tilde{C}\cup C_{3}$ was again updated. The UAV continued to follow this procedure, selecting the next nearest centroid and updating the visited set, until all data have been collected from each subcluster. In this manner, the UAV followed the shortest possible flight path, as shown in Figure 4. After travelling through all *K* subclusters ($\tilde{C}\equiv C$) and collecting all data from wireless sensor members in each subcluster, the UAV’s task was complete and it returned to the mobile base station. For the real-time application of a UAV-assisted WSN, the UAV would repeat the round trips summarized in Table 5.

Observing Table 4, notice that numbers with inclined lines (e.g., 0) and numbers with bold (e.g., **0.5005**) mean visited clusters and next nearest clusters. For clarity, when UAV visited cluster $C_{1}$ (row with $C_{1}$), the UAV selects the next-nearest cluster (i.e., $C_{3}$) and ignores cluster $C_{1}$. Next, the UAV visited cluster $C_{3}$ (row with $C_{3}$), the UAV selects the next-nearest cluster (i.e., $C_{4}$) and ignores clusters $C_{1}$ and $C_{3}$. The remaining rows in Table 4 have the same meaning.
**Algorithm 1***K*-means clustering for the optimal number of subclusters and shortest path determined from a centroid to the next nearest centroid**Input:** 
Generate a wireless sensor network with a number *N* of randomly positioned wireless sensors;**Output:** 
An optimal number of subclusters *K* and subcluster centroids;1:Initialize variables $k_{min}=4$, $k_{max}=\frac{N}{k_{min}}$;2:Attempt $\forall k=\left\{k_{\min},\ldots ,k_{\max}\right\}$ to find the optimal number *K* of subclusters, computed according to Proposition 1;3:Find the centroid positions for *K* subclusters by applying *K*-means clustering;4:Compute the pairwise distances between the mobile base station *B* and subcluster centroids;5:Select the nearest centroid and update $\tilde{C}$;6:**while**$\left|C\cap \tilde{C}\right|\neq \left|C\right|$**do**7:   Compute the pairwise distances between the current centroid and other centroids;8:   Select the nearest centroid and then update $\tilde{C}$.9:**end while**10:**return** the number of subclusters *K*, centroid positions $C_{k}\left(x,y\right)$ for k∈K and the shortest path.

### 2.3. Channel Modelling for a UAV-Assisted WSN

In our previous work [33], we considered free space (i.e., air-to-ground (A2G), ground-to-air (G2A) and air-to-air (A2A)) and first introduced the flat-earth distance based on real latitudes and longitudes. The proposed solutions were effective in determining and tracking the optimal positions for the UAV. A separate study [33] examined the problems related to channel modelling in a WSN which contained multiple subclusters. In the current study, we address the uplinks, i.e., the channels from the wireless sensors to the UAV ($H_{S_{n},U}$) and the channels from the UAV to the mobile base station ($H_{U,B}$). The precoding channel matrices $H_{S_{n},U}$ and $H_{U,B}$ are expressed by
(2)$$\begin{array}{c} H_{S_{n},U}=\left[\begin{array}{ccc} h_{S_{n},U}^{\left(1,1\right)} & \cdots & h_{S_{n},U}^{\left(1,A_{U}\right)}\\ \vdots & \ddots & \vdots \\ h_{S_{n},U}^{\left(A_{S_{n}},1\right)} & \cdots & h_{S_{n},U}^{\left(A_{S_{n}},A_{U}\right)} \end{array}\right]\in C^{A_{S_{n}}\times A_{U}}, \end{array}$$
where $A_{S_{n}}$ and $A_{U}$ are the number of antennae on the wireless sensor $S_{n}$ and UAV *U*, respectively; the channel coefficient $h_{S_{n},U}^{\left(.,.\right)}\in H_{S_{n},U}$ is formulated according to $h_{S_{n},U}^{\left(.,.\right)}=g{\left(d_{S_{n},U}^{G2A}\right)}^{-\varepsilon }$, where *g* is the Rayleigh fading channel, ε is the path-loss exponent, and $d_{S_{n},U}^{G2A}$ is the G2A distance from the sensor node $S_{n}$ to UAV *U*. Note that the free-space distance based on latitude and longitude is given by expression ([33], Equation (2)). For simplicity and without loss of generality, all wireless sensors are allocated with Cartesian coordinates in a three-dimensional space. The G2A distance from wireless sensor $S_{n}$ to UAV *U* is therefore given by $d_{S_{n},U}^{\left(G2A\right)}=\sqrt{{\left|x_{S_{n}}-x_{U}\right|}^{2}+{\left|y_{S_{n}}-y_{U}\right|}^{2}+{\left|z_{S_{n}}-z_{U}\right|}^{2}}$, where the *x*, *y* and *z* axes represent latitude, longitude and altitude, respectively, on a flat earth.

Similarly, the precoding channel matrix $H_{U,B}$ is expressed as
(3)$$\begin{array}{c} H_{U,B}=\left[\begin{array}{ccc} h_{U,B}^{\left(1,1\right)} & \cdots & h_{U,B}^{\left(1,A_{U}\right)}\\ \vdots & \ddots & \vdots \\ h_{U,B}^{\left(A_{B},1\right)} & \cdots & h_{U,B}^{\left(A_{B},A_{U}\right)} \end{array}\right]\in C^{A_{B}\times A_{U}}, \end{array}$$
where $A_{B}$ is the number of antennae at the mobile base station, and the channel coefficient $h_{U,B}^{\left(.,.\right)}\in H_{U,B}$ is formulated according to $h_{U,B}^{\left(.,.\right)}=g{\left(d_{S_{n},U}^{A2G}\right)}^{-\varepsilon }$, where $d_{U,B}^{A2G}$ is the A2G distance from the UAV *U* to the mobile base station *B* and given by $d_{U,B}^{\left(A2G\right)}$ = $\sqrt{{\left|x_{U}-x_{B}\right|}^{2}+{\left|y_{U}-y_{B}\right|}^{2}+{\left|z_{U}-z_{B}\right|}^{2}}$.

### 2.4. UAV Joint Schedule

This study introduces a novel scheduling protocol for a UAV-assisted WSN. The coefficient *t* is the time required to complete a transmission cycle of three phases, i.e., $\lambda_{1}$, $\lambda_{2}$ and $\lambda_{3}$, where $\lambda_{1}$ is the first phase during which the UAV receives signals from the sensor nodes in a cluster, $\lambda_{2}$ is the second phase during which the UAV receives radiofrequency energy from the mobile base station, and $\lambda_{3}$ is the third phase during which the UAV transmits the superimposed signals to the mobile base station for data analysis. Figure 5 depicts an electronic control unit (ECU) which performs a task corresponding to a predefined operation in a common UAV schedule.

The ECU implements an electronic switch which applies three successive modes during the same transmission block *t*:In phase $\lambda_{1}$, the interface of the receiving signal circuit is active while the other interfaces are inactive. The UAV receives signals from the sensor nodes in the currently visited subcluster, given by (5).In phase $\lambda_{2}$, the interface of the EH circuit is active while the other interfaces are inactive. The UAV receives radiofrequency energy from the mobile base station *B*, given by (10), while the ECU decodes the messages from the signals received from the wireless sensors.In phase $\lambda_{3}$, the interface of the transmitting signal circuit is active while the other interfaces are inactive. The UAV encodes the messages received in the first phase and forwards the superimposed signals to the mobile base station *B*, given by (11).

#### 2.4.1. Phase 1: Uplinks between Wireless Sensors and the UAV

In the first phase, having selected the next nearest subcluster centroid, the UAV visits and hovers at the selected centroid and receives signals wirelessly from the subcluster’s sensors. Figure 6 illustrates the procedure of receiving signals and processing data at the UAV during the first phase.

Based on the number of subclusters *K* and the global period *T*, we calculate the UAV period *t* as follows:(4)$$\begin{array}{c} t^{}=\left\{\begin{array}{c} mod\left\{T,K\right\},\;s.t.\;mod\left\{T,K\right\}>0,\\ K,\;\;\;s.t.\;mod\left\{T,K\right\}=0, \end{array}\right. \end{array}$$
where the $mod\left\{T,K\right\}$ function refers to the modulo value between *T* and *K* (e.g., for T=10 and K=4, $t^{}=mod\left\{T,K\right\}=2$). At each UAV period *t*, the UAV selects the next nearest centroid using the centroid-to-next-nearest-centroid algorithm. According to the trajectory mapped in Table 5, for T=10, K=4 and $t^{}=2$, the UAV selects the subcluster $C_{3}$ and serves wireless sensors $S_{1}^{\left(1,3\right)}$, $S_{6}^{\left(2,3\right)}$, $S_{8}^{\left(3,3\right)}$, $S_{12}^{\left(4,3\right)}$, $S_{14}^{\left(5,3\right)}$, $S_{15}^{\left(6,3\right)}$, $S_{17}^{\left(7,3\right)}$, $S_{18}^{\left(8,3\right)}$, $S_{25}^{\left(9,3\right)}$, $S_{26}^{\left(10,3\right)}$, $S_{33}^{\left(11,3\right)}$, $S_{34}^{\left(12,3\right)}$ and $S_{36}^{\left(13,3\right)}$. The signals received from the wireless sensors in subcluster $C_{k}$ over UAV period *t* are given by
(5)$$\begin{array}{cc} \; & \max_{\left[A_{S_{n}}\times A_{U}\right]}\left\{y_{S_{n}^{\left(i,k\right)}}^{}\left(T,N,K\right)\right\}=\sqrt{P_{S_{n}}}\max_{\left[A_{S_{n}}\times A_{U}\right]}\left\{\left|H_{S_{n},U}^{}\right|\right\}x_{S_{n}^{\left(i,k\right)}}+n_{U}, \end{array}$$
(6)$$\begin{array}{cc} \; & s.t.\;1\leq n\leq N,1\leq k\leq K,1\leq i\leq N_{k},N_{k}=\left|C_{k}\right|, \end{array}$$
where *t* is obtained from (4), *k* is mapped as in Table 5, and $P_{S_{n}}$ is the transmit power of sensor $S_{n}$. For simplicity, we assume that $P_{S_{1}}=\ldots =P_{S_{N}}$.

Let us denote $D\left(T,N,K\right)$, which is the mathematical description of a diagonal matrix, as follows:(7)$$\begin{array}{c} D\left(T,N,K\right)=diag{\left(1,\ldots ,1\right)}_{N_{k}\times N_{k}}={\left[\begin{array}{ccc} 1 & & \\ & \ddots & \\ & & 1 \end{array}\right]}_{\left|C_{k}\right|\times \left|C_{k}\right|}\;, \end{array}$$
where the diagonal matrix D has the size $\left|C_{k}\right|\times \left|C_{k}\right|$ for transmission period *T* and all nondiagonal elements are zero, as indicated in Figure 6. The predecoded matrix obtained at the UAV is derived by multiplying the received signal matrix (5) with the diagonal matrix (7); thus, $preDecode\left(T,N,K\right)=y_{S_{n}^{\left(i,k\right)}}^{}\left(T,N,K\right)\times D\left(T,N,K\right)$. The UAV selects each element in the predecoded matrix to obtain the SINR at the point when the UAV decodes message $x_{S_{n}}$ from sensor $S_{n}\in C_{k}$, as follows:(8)$$\begin{array}{c} \max_{\left[A_{S_{n}}\times A_{U}\right]}\left\{\gamma_{U-x_{S_{n}^{\left(i,k\right)}}}^{}\left(T,N,K\right)\right\}=\rho_{S_{n}}\max_{\left[A_{S_{n}}\times A_{U}\right]}\left\{{\left|H_{S_{n},U}\right|}^{2}\right\}, \end{array}$$
where the signal-to-noise ratio (SNR) ρSn=PSnPSnN0N0. For simplicity, we assume that $\rho_{S_{1}}=\ldots =\rho_{S_{N}}$.

We then obtain the instantaneous bit rate at the point when the UAV decodes message $x_{S_{n}^{\left(i,k\right)}}$ from sensor $S_{n}^{\left(i,k\right)}$, as follows:(9)$$\begin{array}{c} \max_{\left[A_{S_{n}}\times A_{U}\right]}\left\{R_{U-x_{S_{n}^{\left(i,k\right)}}}^{}\left(T,N,K\right)\right\}=\frac{1}{2}\log_{2}\left(1+\max_{\left[A_{S_{n}}\times A_{U}\right]}\left\{\gamma_{U-x_{n}^{\left(i,k\right)}}^{}\left(T,N,K\right)\right\}\right). \end{array}$$

#### 2.4.2. Phase 2: Prolong the UAV’s Online Time with EH

The most challenging aspect of deploying a UAV is managing its power limitations as a small, lightweight aircraft. We propose applying SWIPT techniques to prolong the UAV’s online time. In a previous study [33] (Figure 4), we adopted a power splitting protocol. In the current study, we applied a time-switching technique for its advantages in a WSN (Figure 5); the technique is different from the proposed time-switching models in [33] (Figure 4). In phase $\lambda_{2}$, the UAV harvests radiofrequency energy from the mobile base station according to
(10)$$\begin{array}{c} EH\left(T,N,K\right)=\eta P_{B}\sigma_{B,U}, \end{array}$$
where $P_{B}$ is the power domain at the mobile base station, and $\sigma_{B,U}$ is the expected channel gain between the mobile base station and the UAV at its current UAV position. It is important to note that η is the collected energy factor and that we assume η=1 for simplicity.

#### 2.4.3. Phase 3: Transmitting Signals

In [10], the authors applied the amplify-and-forward protocol at the UAV to receive and forward signals to a single device. In the current study, we implemented a decode-and-forward protocol at the UAV to ensure that the UAV received, decoded and encoded messages successfully before forwarding the superimposed signals to the mobile base station. To improve latency, we applied the emerging NOMA technique for its high spectral efficiency. The UAV *U* encoded the messages $\forall x_{{S_{n}}^{\left(i,k\right)}}\in X_{k}$ from the sensors in the current subcluster $C_{k}$ and superimposed them into the signal by sharing the power domain $P_{U}$ and using different power allocation factor $\alpha_{S_{n}}^{\left(i,k\right)}$. From the precoding matrix $H_{U,B}$, as given by (3), only the best channel was selected for signal transmission.

In the third phase $\lambda_{3}$ of transmission block *t*, the mobile base station *B* received radiofrequency signals as follows:(11)$$\begin{array}{c} \max_{\left[A_{U}\times A_{B}\right]}\left\{y_{B}^{}\left(T,N,K\right)\right\}=\max_{\left[A_{U}\times A_{B}\right]}\left\{\left|H_{U,B}\right|\right\}\sum_{\forall S_{n}^{\left(i,k\right)}\in C_{k}}^{}\sqrt{P_{U}\alpha_{S_{n}^{\left(i,k\right)}}}x_{S_{n}^{\left(i,k\right)}}+n_{B}, \end{array}$$
where $n_{B}$ is the AWGN (i.e., $n_{B}∼CN\left(0,N_{0}\right)$ with zero mean and variance $N_{0}$) at the mobile base station *B*, $P_{U}$ is the power domain at UAV *U*, and $\alpha_{S_{n}^{\left(i,k\right)}}$ is the power allocation factor for message $x_{S_{n}^{\left(i,k\right)}}$ of wireless sensor $S_{n}^{\left(i,k\right)}$. The NOMA technique applies superimposed coding by sharing the power domain and therefore, the power allocation strategy strongly affects the success or failure of decoding a message. Previous studies [17,21,34] have also applied power allocation strategies; the current study, however, addresses a WSN divided into multiple subclusters and therefore proposes the novel power allocation strategy described below.

**Proposition** **3.**
*The power allocation strategy for transmitting messages from wireless sensors over UAV transmission period t while the UAV visits subcluster $C_{k}$ is expressed as follows:*

(12)
$$\begin{array}{c} \alpha_{S_{n}^{\left(i,k\right)}}=\frac{N_{k}-i+1}{\sum_{j=1}^{N_{k}}j}, \end{array}$$

*where a sensor with a higher priority is allocated a larger power allocation factor; for example, sensor $S_{1}$, which has the highest priority and is the first member in subcluster $C_{3}$, is allocated the largest power allocation factor $\alpha_{S_{1}^{\left(1,3\right)}}=0.1428$, whereas sensor $S_{36}$, which has the lowest priority and is the last member in the subcluster $C_{3}$, is allocated the smallest power allocation factor $\alpha_{S_{36}^{\left(13,3\right)}}=0.011$. For clarity, we applied the power allocation factors presented in Table 6. From Equation (12), the power allocation strategy in the subcluster is constrained such that $\alpha_{S_{n}^{\left(i,k\right)}}>\ldots >\alpha_{S_{n}^{\left(1,k\right)}}$ and $\alpha_{S_{n}^{\left(i,k\right)}}+\ldots +\alpha_{S_{n}^{\left(1,k\right)}}=1$.*


The SINR at the mobile base station *B* when *B* decodes message $x_{S_{n}^{\left(i,k\right)}}\in X_{k}$ treats other messages $x_{S_{n}^{\left(j,k\right)}}\in X_{k}$, where $\alpha_{S_{n}^{\left(j,k\right)}}<\alpha_{S_{n}^{\left(i,k\right)}}$, and AWGN $n_{B}$ as interference by applying SIC: (13)$$\begin{array}{cc} \max_{\left[A_{U}\times A_{B}\right]}\left\{\gamma_{B-x_{S_{n}^{\left(i,k\right)}}}^{}\left(T,N,K\right)\right\}\; & =\frac{\max_{\left[A_{U}\times A_{B}\right]}\left\{{\left|H_{U,B}\right|}^{2}\right\}\alpha_{S_{n}^{\left(i,k\right)}}\rho_{U}\sigma_{U,B}}{\max_{\left[A_{U}\times A_{B}\right]}\left\{{\left|H_{U,B}\right|}^{2}\right\}\rho_{U}\sigma_{U,B}\sum_{j=i+1}^{N_{k}}\alpha_{S_{n}^{\left(j,k\right)}}+1}, \end{array}$$(14)$$\begin{array}{cc} \; & =\max_{\left[A_{U}\times A_{B}\right]}\left\{{\left|H_{U,B}\right|}^{2}\right\}\alpha_{S_{n}^{\left(N_{k},k\right)}}\rho_{U}\sigma_{U,B}, \end{array}$$
where $i<N_{k}$ in (13) and $i=N_{k}$ in (14).

The maximum instantaneous bit-rate threshold attained if the mobile base station decodes message $x_{S_{n}^{\left(i,k\right)}}$ in the best-received signal, given by (11), is expressed as:(15)$$\begin{array}{c} \max_{\left[A_{U}\times A_{B}\right]}\left\{R_{B-x_{S_{n}^{\left(i,k\right)}}}^{}\left(T,N,K\right)\right\}=\frac{1}{2}\log_{2}\left(1+\max_{\left[A_{U}\times A_{B}\right]}\left\{\gamma_{B-x_{S_{n}^{\left(i,k\right)}}}^{}\left(T,N,K\right)\right\}\right), \end{array}$$
where $\forall x_{S_{n}^{\left(i,k\right)}}\in X_{k}$, and $\forall i=\left\{1,\ldots ,N_{k}\right\}$.

## 3. System Performance Analysis

In this section, we derive the novel closed-form expressions for the independent outage probability at the UAV and the dependent outage probability at the mobile base station.

### 3.1. Outage Probability Performance at the UAV

**Theorem** **1.**
*The independent outage probability at the UAV U relates to the UAV’s unsuccessful decoding of the message in the received signal, given by (5). In other words, the maximum instantaneous bit-rate threshold, given by (9), cannot reach the predefined bit-rate threshold R. The independent outage probability at the UAV U in transmission block t is therefore expressed as*

(16)
$$\begin{array}{c} OP_{U-x_{S_{n}^{\left(i,k\right)}}}^{}\left(T,N,K\right)=1-\mathrm{Pr}\left\{\max_{\left[A_{S_{n}}\times A_{U}\right]}\left\{R_{U-x_{S_{n}^{\left(i,k\right)}}}^{}\left(T,N,K\right)\right\}\geq R\right\}. \end{array}$$



Based on Equation (16), we propose Algorithm 2 to calculate the Monte Carlo simulations for the outage probability at the UAV *U*.
**Algorithm 2** Calculate the outage probability at the UAV *U* from (16) for transmission block *t***Input:** 
Initialize the parameters as in Table 1 and randomly generate $10^{6}$ samples of each fading channel over a Rayleigh distribution**Output:** 
Simulate (Sim) the results for outage probability at the UAV *U* in transmission block *t*1:**for**k=1 to the optimal number of subclusters *K* **do**2:   **for** i=1 to the number $N_{k}$ of sensor members within the subcluster $C_{k}$ **do**3:     Calculate the SINR at the UAV from (8);4:     Calculate the achievable maximum instantaneous bit-rate from (9);5:     Initialize variable count←0;6:     **for** l=1 to $10^{6}$ samples **do**7:        **if** $\left(\min_{S_{n}^{\left(i,k\right)}\in C_{k}}\max_{\left[A_{S_{n}}\times A_{U}\right]}\left\{R_{S_{n}^{\left(i,k\right)}}^{}\left(T,N,K\right)\right\}\geq R\right)$ **then**8:          count←count+1;9:        **end if**10:     **end for**11:     $OP_{U-x_{S_{n}}^{\left(i,k\right)}}^{}\left(T,N,K\right)=1-\frac{count}{10^{6}}$;12:   **end for**13:   $OP_{U}^{}\left(T,N,K\right)=\frac{1}{N_{k}}\sum_{k=1}^{N_{k}}OP_{U-x_{S_{n}}^{\left(i,k\right)}}^{}\left(T,N,K\right)$;14:**end for**15:**return** Outage probabilities at UAV $OP_{U}^{}\left(T,N,K\right)$;

**Remark** **2.**
*From expression (16), we obtain the outage probability at the UAV over Rayleigh distributions:*

(17)
$$\begin{array}{c} OP_{U-x_{S_{n}^{\left(i,k\right)}}}^{}\left(T,N,K\right)=\sum_{\psi =0}^{A_{S_{n}}A_{U}}\frac{{\left(-1\right)}^{\psi }\left(A_{S_{n}}A_{U}\right)!}{\psi !\left(A_{S_{n}}A_{U}-\psi \right)!}\exp\left(-\frac{\psi \gamma }{\rho_{S_{n}}\sigma_{S_{n},U}}\right), \end{array}$$

*where the SINR threshold is given by $\gamma =2^{2R}-1$. It is important to note that Equation (17) obtains the independent outage probabilities at the UAV. Generally, the outage probability at the UAV is calculated from $OP_{U}^{}\left(T,N,K\right)=\frac{1}{N_{k}}\sum_{i=1}^{N_{k}}OP_{U-x_{S_{n}^{\left(i,k\right)}}}^{}\left(T,N,K\right)$.*


See Appendix C for the proof.

### 3.2. Outage Probability at the Mobile Base Station

**Theorem** **2.**
*The dependent outage event at the mobile base station occurs when the flying relay (FR)-UAV either cannot decode at least the message $x_{S_{n}^{\left(i,k\right)}}\in X_{k}$ or the mobile base station B cannot decode at least the message $x_{S_{n}^{\left(i,k\right)}}\in X_{k}$ from the best-received signal $y_{B}^{}\left(T,N,K\right)$, given by (11). The outage probability at the mobile base station with an underlying U-assisted multi-input multioutput (MIMO)-NOMA network is therefore expressed as*

(18)
$$\begin{array}{cc} OP_{B}^{}\left(T,K,N\right)=1- & \mathrm{Pr}\left\{\min_{x_{S_{n}^{\left(i,k\right)}}\in X_{k}}\max_{\left[A_{S_{n}}\times A_{U}\right]}\left\{R_{U-x_{S_{n}^{\left(i,k\right)}}}^{}\left(T,N,K\right)\right\}\geq R,\right.\\ & \left.\min_{x_{S_{n}^{\left(i,k\right)}}\in X_{k}}\max_{\left[A_{U}\times A_{B}\right]}\left\{R_{B-x_{S_{n}^{\left(i,k\right)}}}^{}\left(T,N,K\right)\right\}\geq R\right\}. \end{array}$$



Based on Equation (18), we propose Algorithm 3 to calculate the Monte Carlo simulations for outage probability at the mobile base station for transmission block *t* over Rayleigh distributions.
**Algorithm 3** Calculate the outage probability at the mobile base station from (18) for transmission block *t* over Rayleigh distributions**Input:** 
Initialize the parameters as in Table 1 and randomly generate $10^{6}$ samples of each fading channel over a Rayleigh distribution;**Output:** 
Simulate (Sim) the results for outage probability at the mobile base station *B*;1:**for**k=1 to the optimal number *K* of the subcluster **do**2:   **for** i=1 to the number of sensors $N_{k}$ **do**3:     Calculate the SINR at the UAV *U* from (8);4:     Calculate the achievable maximum instantaneous bit-rate at the UAV *U* from (9);5:     Calculate the minimum-maximum instantaneous bit-rate threshold at the UAV *U* from (9);6:     Calculate the SINR at the mobile base station from (13) or (14);7:     Calculate the achievable maximum bit-rate at the mobile base station from (15);8:     Calculate the achievable minimum-maximum bit-rate at the mobile base station from (15);9:     Initialize variable count←0;10:     **for** l=1 to $10^{6}$ samples **do**11:        **if** $\left(\min\left\{\min_{x_{S_{n}^{\left(i,k\right)}}\in X_{k}}\max_{\left[A_{S}\times A_{U}\right]}\left\{R_{U-x_{S_{n}^{\left(i,k\right)}}}^{}\left(T,N,K\right)\right\}\right.\right.$, $\;\;\;\left.\left.\min_{x_{S_{n}^{\left(i,k\right)}}\in X_{k}}\max_{\left[A_{U}\times A_{B}\right]}\left\{R_{B-x_{S_{n}^{\left(i,k\right)}}}\left(T,N,K\right)\right\}\right\}\geq R\right)$
**then**12:          count←count+1;13:        **end if**14:     **end for**15:     $OP_{B-x_{S_{n}}^{\left(i,k\right)}}^{}\left(T,N,K\right)=1-\frac{count}{10^{6}}$;16:   **end for**17:   $OP_{B}^{}\left(T,N,K\right)=\frac{1}{N_{k}}\sum_{k=1}^{N_{k}}OP_{B-x_{S_{n}}^{\left(i,k\right)}}^{}\left(T,N,K\right)$;18:**end for**19:**return** Dependent outage probability at the mobile base station $OP_{B}^{}\left(T,N,K\right)$;

**Remark** **3.**
*The outage probability at the mobile base station in transmission block t is given by (18) from Theorem 1 and expressed in novel closed-form as follows:*

(19)
$$\begin{array}{cc} \; & \begin{array}{c} OP_{B}^{\left(i\right)}\left(T,N,K\right)=\max\left\{\sum_{\psi =0}^{A_{S}A_{U}}\frac{{\left(-1\right)}^{\psi }\left(A_{S}A_{U}\right)!}{\psi !\left(A_{S}A_{U}-\psi \right)!}\exp\left(-\frac{\psi \gamma_{}}{\min\left\{\rho_{S_{n}}\sigma_{S_{n},U}^{}\right\}}\right),\right.\\ \;\;\;\;\;\left.\sum_{\psi =0}^{A_{U}A_{B}}\frac{{\left(-1\right)}^{\psi }\left(A_{U}A_{B}\right)!}{\psi !\left(A_{U}A_{B}-\psi \right)!}\exp\left(-\frac{\psi \gamma_{}}{\beta \rho_{U}\sigma_{U,B}^{}}\right)\right\} \end{array}\;, \end{array}$$


(20)
$$\begin{array}{cc} s.t.\; & \beta_{i}=\alpha_{S_{n}^{\left(i,k\right)}}-\gamma \sum_{j=i+1}^{N_{k}}\alpha_{S_{n}^{\left(j,k\right)}}, \end{array}$$


(21)
$$\begin{array}{cc} \; & \beta =\min_{i=\left\{1,\ldots ,N_{k}\right\}}\left\{\beta_{i}\right\}, \end{array}$$

*where SINR threshold $\gamma =2^{2R}-1$.*


See Appendix D for the proof.

## 4. Numerical Results and Discussion

In this section, we examine the individual WSN and discuss the results of the study. For the purposes of the analysis and the Monte Carlo simulations, a random number of wireless sensors *N* was generated and randomly distributed according to the positions illustrated in Figure 1. Unless specified otherwise, we assumed that the mobile base station’s position was at coordinate $B\left(0,0,0\right)$ and that the UAV’s position $U\left(x,y,1\right)$ determined by *K*-means clustering had a fixed altitude at z=1. The number of antennae equipped at the wireless sensors, UAV and mobile base station was $A_{S_{n}}=A_{U}=A_{B}=2$. The *K*-means algorithm determined the optimal number of subclusters as K=4. The path-loss exponent factor was ε=4. The list of pairwise distances from each subcluster centroid to the mobile base station was $d_{C_{1},B}^{A2G}=0.494$, $d_{C_{2},B}^{A2G}=0.8788$, $d_{C_{3},B}^{A2G}=0.9268$ and $d_{C_{4},B}^{A2G}=1.1620$. The nearest subcluster to the mobile base station was therefore $C_{1}$. The UAV selected subcluster $C_{1}$ for the global period $T=\left\{1,5,9,13,\ldots \right\}$. At global period $T=\left\{2,6,10,14,\ldots \right\}$, the UAV then selected the next nearest subcluster to its current subcluster $C_{1}$, i.e., subcluster $C_{3}$, because distances $d_{C_{1},C_{3}}^{A2A}=0.5005<d_{C_{1},C_{2}}^{A2A}=0.5602<d_{C_{1},C_{4}}^{A2A}=0.7195$. At global period $T=\left\{3,7,11,15,\ldots \right\}$, the UAV again selected the next nearest subcluster to subcluster $C_{3}$, i.e., subcluster $C_{4}$, since distances $d_{C_{3},C_{4}}^{A2A}=0.4501<d_{C_{3},C_{4}}^{A2A}=0.7166$. At global period $T=\left\{4,8,12,17,\ldots \right\}$, the UAV selected the next nearest subcluster to subcluster $C_{4}$, i.e., subcluster $C_{2}$, because the final distances $d_{C_{4},C_{2}}^{A2A}=0.5262$. The UAV thus selected the shortest trajectory $C_{1}\to C_{3}\to C_{4}\to C_{2}$. Without loss of generality and for simplicity, we assumed that $\lambda_{1}=\lambda_{2}=\lambda_{3}=\frac{t}{3}$ for a single round trip of the UAV and global period $T=\left\{1,2,3,4\right\}$.

### 4.1. Numerical Results

Figure 7a–d plot the outage probabilities at the UAV at the point when it decoded the received signals from wireless sensors in subclusters $C_{1}$, $C_{3}$, $C_{4}$ and $C_{2}$, respectively. The bit-rate threshold for all wireless sensors was R=1.5 bps/Hz. The outage probabilities at the majority of wireless sensors were very similar as SNR $\rho_{S_{n}}\to \infty$; however, the graphs in Figure 7a indicate that the outage probability of sensor $S_{23}$ was worse than the outage probability at the other sensors of the same subcluster $C_{1}$. Figure 3 indicates that wireless sensor $S_{23}$ was the farthest from the subcluster centroid $C_{1}$ ($d_{S_{23}^{\left(7,1\right)}}^{G2A}=1.0479$). The results verified the efficiency of the *K*-means algorithm. It is important that the bit-rate threshold R for the wireless sensors was set to R=1.5 bps/Hz. However, the UAV successfully decoded most of the messages from the wireless sensors, achieving a high outage probability performance (Figure 7). We conclude that the outage probability performance of the majority of wireless sensors in each subcluster was equal since they were evenly distributed around the subcluster’s centroid. The Monte Carlo simulations given by (16) were also verified by the analysis results given by (17).

Next, we examined the results for the mobile base station and obtained its outage probability performance at the points when the UAV visited subclusters $C_{1}$, $C_{3}$, $C_{4}$ and $C_{2}$ and forwarded the superimposed signals, given by (11), to the base station (Figure 8a–d). The outage probability performance of the mobile base station was poorer than the outage probability performance at the UAV (Figure 7a–d), even though the bit-rate threshold was set to R=0.1 bps/Hz. This may have been because the UAV was deployed with NOMA and therefore, the sensors were forced to share the power domain to transmit the messages in the superimposed signal. This means that the last member in the subcluster was allocated a very small power allocation factor, given by (12). These power allocation factors are presented in Table 6. Subcluster $C_{3}$ contained $N_{3}=13$ wireless sensors, and the last wireless sensor in $C_{3}$ ( $S_{36}^{\left(13,3\right)}$) was allocated the lowest power allocation factor ($\alpha_{S_{36}^{\left(13,3\right)}}$). Therefore, despite a global optimization of the subclusters and the positions of the cluster centroids by the *K*-means algorithm, the large number of wireless sensors in the subcluster unfortunately led to unsatisfactory results.

### 4.2. Discussion

The outage probability performance at the mobile base station was strongly affected by several factors, such as the UAV’s transmit power $P_{U}$, the distance of the UAV from the mobile base station $d_{U,B}$ and the number $N_{k}$ of messages transmitted in the superimposed signals. The UAV was not able to increase the transmit power $P_{U}$, however, because of its power limitations. A large number of antennae at both the wireless sensors and the UAV could not be equipped since the constraints for a small size, light weight and low cost did not permit it. It was also not possible to reduce the distance from the UAV to the mobile base station because of obstructions in the terrain. To address these conditions, we equipped a larger number of antennae at the mobile base station, as the mobile base station incorporated a generator and energy was not a significant problem. Therefore, equipping $A_{B}=32$ antennae at the mobile base station instead of the same number at the UAV $A_{S_{n}}=A_{U}=A_{B}=2$ (Figure 8a–d) yielded the results in Figure 9a–d. It is clear that the outage probability improved significantly at the mobile base station for $A_{B}=32$ while $A_{S_{n}}=A_{U}=2$. It is also clear that the outage probability performance at the mobile base station when the UAV visited subcluster $C_{1}$ improved greatly since this subcluster $C_{1}$ was closer than the other subclusters. The outage probabilities at the other subclusters also improved as the SNR $\rho_{U}\to \infty$.

## 5. Conclusions

This study presented a general WSN containing a randomly distributed number of wireless sensors with three-dimensional Cartesian coordinates. To improve the WSN’s performance, we applied a *K*-means algorithm and gap statistic method to optimize sensor clustering into a number of subclusters *K*. The UAV’s trajectory was calculated with an algorithm which determined the shortest path between the subcluster centroids. The aims of the study were achieved (i.e., flexible deployment, low cost and high reliability) through the effective proposed solutions, and the results were verified with both Monte Carlo simulations and theoretical analysis. Although the study provided some benefits from the application of the *K*-means algorithm for wireless sensor clustering, some problems still persisted that can be studied in future work. Future studies can investigate the problems with (1) fragmented power resources created by an imbalance in the number of subcluster sensors and (2) some clusters covering a larger geographic area than others as a result of sparsely distributed sensors. As a potential solution, we propose dividing the network into larger clusters when the number of sensors reaches a certain threshold.

## Figures and Tables

**Figure 1 sensors-23-02345-f001:**
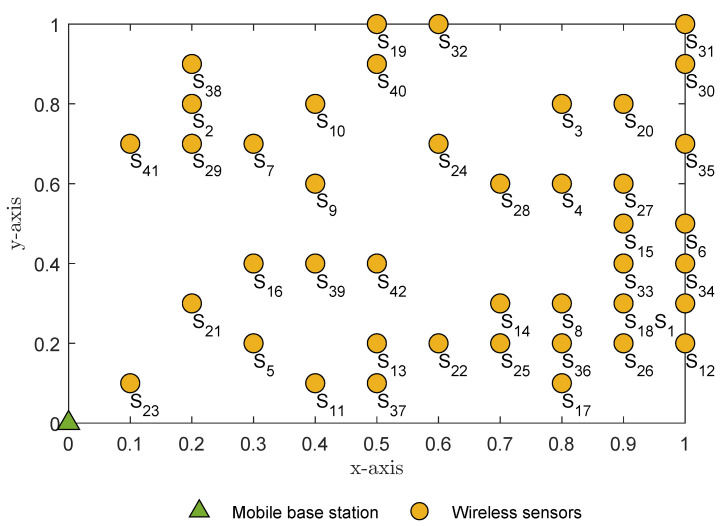
Random positioning of sensor nodes, where 20≤N≤50.

**Figure 2 sensors-23-02345-f002:**
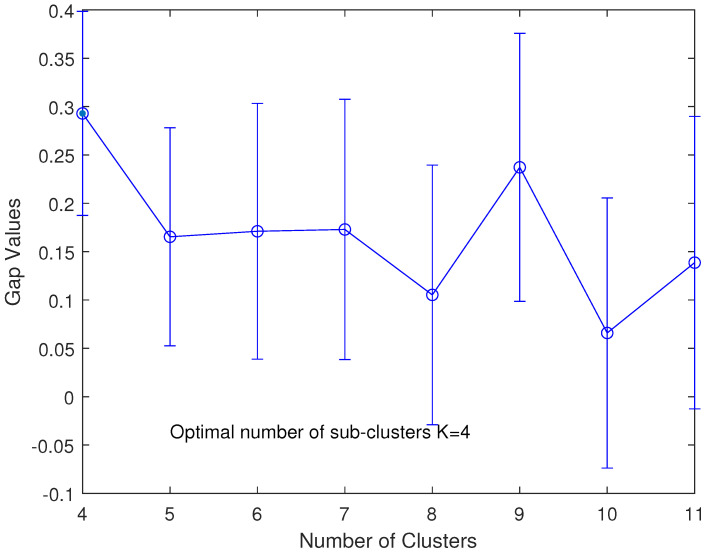
Optimized number of subclusters using the gap statistic method, the optimal number of clusters at K=4 satisfying the first maximum standard error.

**Figure 3 sensors-23-02345-f003:**
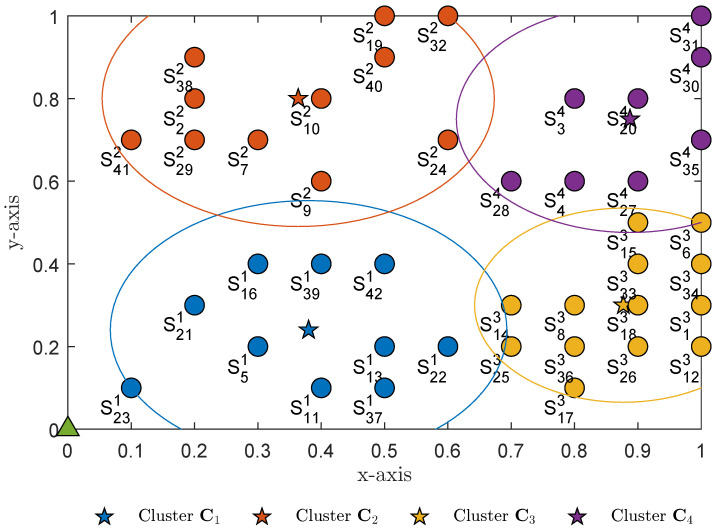
Sensor clustering using the *K*-means algorithm, with optimal number of clusters K=4.

**Figure 4 sensors-23-02345-f004:**
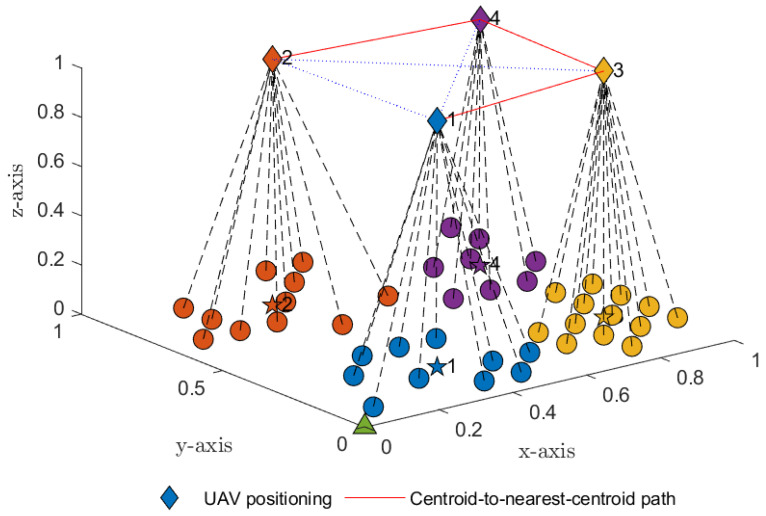
Joint UAV trajectory and the shortest path based on the centroid-to-next-nearest-centroid distance given by Algorithm 1 (i.e., $C_{1}\to C_{3}\to C_{4}\to C_{2}$).

**Figure 5 sensors-23-02345-f005:**
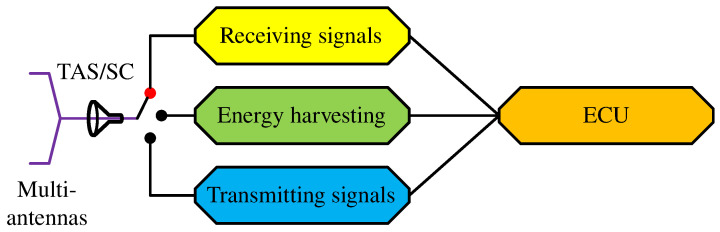
Joint schedule.

**Figure 6 sensors-23-02345-f006:**
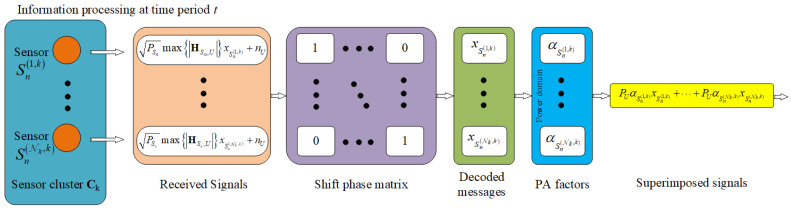
Procedure of processing data at the UAV.

**Figure 7 sensors-23-02345-f007:**
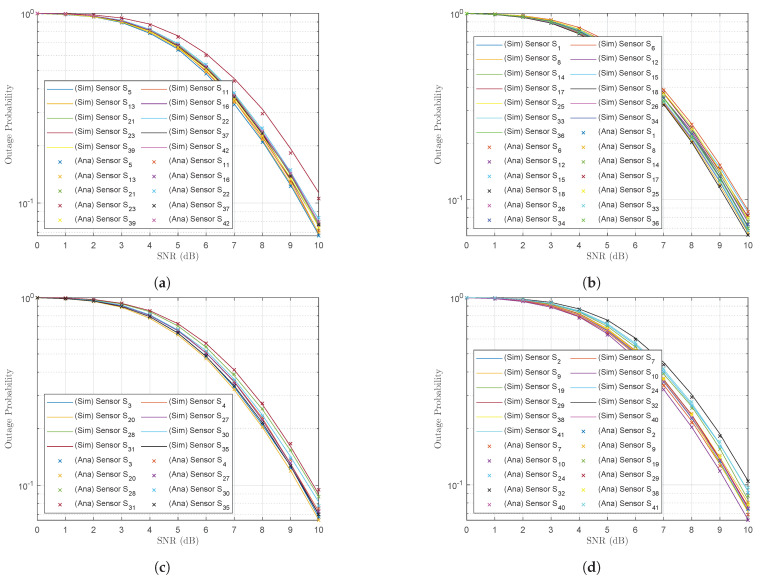
Outage probability at the UAV for the UAV’s subcluster trajectory sequence (**a**) $C_{1}$, (**b**) $C_{3}$, (**c**) $C_{4}$ and (**d**) $C_{2}$.

**Figure 8 sensors-23-02345-f008:**
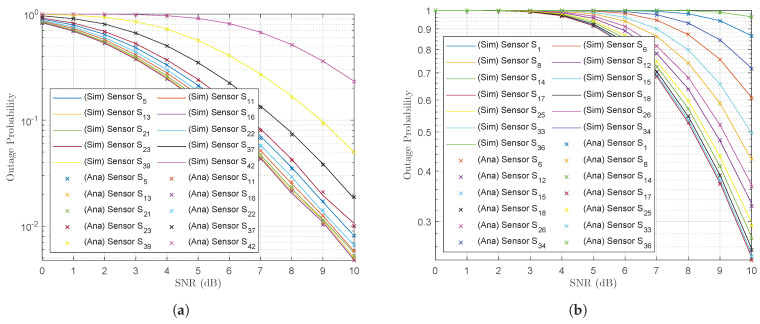
Outage probability at the mobile base station for the UAV’s subcluster trajectory sequence (**a**) $C_{1}$, (**b**) $C_{3}$, (**c**) $C_{4}$ and (**d**) $C_{2}$.

**Figure 9 sensors-23-02345-f009:**
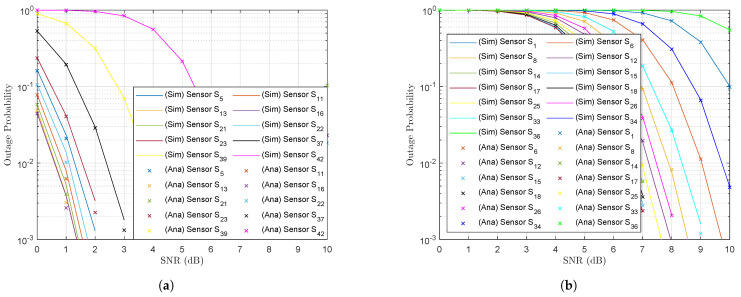
Improved outage probability at the mobile base station equipped with $A_{B}=32$ antennae for the UAV trajectory subcluster sequence (**a**) $C_{1}$, (**b**) $C_{3}$, (**c**) $C_{4}$ and (**d**) $C_{2}$.

**Table 1 sensors-23-02345-t001:** List of important notations.

Notations	Describe	Conditions
*N*	Random number of sensors	20≤N≤50
*K*	Optimal number of clusters given by the *K*-means algorithm, where the number of subclusters *K* is optimized	$k_{min}\leq K\leq \frac{N}{k_{min}}$
$S_{n}$	*n*th sensor node, where a lower value for *n* has higher priority	$n=\left\{1,\ldots ,N\right\}$
C	Global wireless sensor cluster	$C⊃S_{n}\;\;s.t.\;\forall n\in N$, $\left|C\right|=N$
$C_{k}$	*k*th subcluster	$k=\left\{1,\ldots ,K\right\}$, $N_{k}=\left|C_{k}\right|$
$S_{n}^{\left(i,k\right)}$	*n*th sensor is *i*th member of the *k*th subcluster, where a lower value for *i* has higher priority	$i=\left\{1,\ldots ,N_{K}\right\}$, $N_{K}=\left|C_{k}\right|$
*T*	Global transmission time period	$T\in Z^{+}$
*t*	UAV time period	$t=mod\left(T,K\right)\vee K$
$A_{S_{n}},A_{U},A_{B}$	Number of antennae at the sensors, UAV and mobile base station	$A_{S_{n}}\geq 1$, $A_{U}\geq 1$, and $A_{B}\geq 1$
ε	Path-loss exponent factor	ε≥2
$\tilde{C}$	Visited cluster set	Updated after the UAV visits the centroid of a subcluster $C_{k}$ as given by $\tilde{C}\leftarrow \tilde{C}\cup C_{k}$
$H_{S_{n},U}$, $H_{U,B}$	Precoding fading channel matrices from sensors to the UAV and from the UAV to the mobile base station	$H_{S_{n},U}\in C^{A_{S_{n}}\times A_{U}}$ and $H_{U,B}\in C^{A_{U}\times A_{B}}$ have sizes of $A_{S_{n}}\times A_{U}$ and $A_{U}\times A_{B}$, respectively
$\sigma_{S_{n},U}$, $\sigma_{U,B}$	Channel gains	$\sigma_{S_{n},U}=E\left\{{\left|h_{S_{n},U}^{\left(.,.\right)}\right|}^{2}\right\}$, $\sigma_{U,B}=E\left\{{\left|h_{U,B}^{\left(.,.\right)}\right|}^{2}\right\}$
$\alpha_{S_{n}^{\left(i,k\right)}}$	Power allocation factor for sensor $S_{n}$, indexed *i*th in subcluster $C_{k}$	$\alpha_{S_{n}^{\left(1,k\right)}}+\ldots +\alpha_{S_{n}^{\left(N_{k},k\right)}}=1$ and $\alpha_{S_{n}^{\left(1,k\right)}}>\ldots >\alpha_{S_{n}^{\left(N_{k},k\right)}}$
$P_{S_{n}},P_{U},P_{B}$	Respective power domains at the sensors, UAV and mobile base station *B*	Let $P_{S_{1}}=\ldots =P_{S_{N}}$ dB
R	Predefined bit-rate threshold for sensors	bps/Hz
$\gamma_{U-x_{S_{n}^{\left(i,k\right)}}}$, $\gamma_{B-x_{S_{n}^{\left(i,k\right)}}}$	SINR reached at UAV *U* and *B* when message $x_{S_{n}^{\left(i,k\right)}}$ of sensor $S_{n}$ is decoded	SIC decodes the message with the biggest power allocation factor by treating other messages and AWGN as interference
$R_{U-x_{S_{n}^{\left(i,k\right)}}}$, $R_{B-x_{S_{n}^{\left(i,k\right)}}}$	Instantaneous bit rate reached at UAV *U* and mobile base station *B* when message $x_{S_{n}^{\left(i,k\right)}}$ of sensor $S_{n}$ is decoded	bps/Hz
$OP_{U}$, $OP_{B}$	Outage probabilities at UAV *U* and mobile base station *B*	$0\leq OP_{U}\leq 1$, $0\leq OP_{B}\leq 1$, a lower outage probability result is better performance

**Table 2 sensors-23-02345-t002:** Wireless sensor positions are distributed randomly.

Sensors	*x*-Coordinate	*y*-Coordinate	Sensors	*x*-Coordinate	*y*-Coordinate
$S_{1}$	1	0.3	$S_{2}$	0.2	0.8
$S_{3}$	0.8	0.8	$S_{4}$	0.8	0.6
$S_{5}$	0.3	0.2	$S_{6}$	1	0.5
$S_{7}$	0.3	0.7	$S_{8}$	0.8	0.3
$S_{9}$	0.4	0.6	$S_{10}$	0.4	0.8
$S_{11}$	0.4	0.1	$S_{12}$	1	0.2
$S_{13}$	0.5	0.2	$S_{14}$	0.7	0.3
$S_{15}$	0.9	0.5	$S_{16}$	0.3	0.4
$S_{17}$	0.8	0.1	$S_{18}$	0.9	0.3
$S_{19}$	0.5	1	$S_{20}$	0.9	0.8
$S_{21}$	0.2	0.3	$S_{22}$	0.6	0.2
$S_{23}$	0.1	0.1	$S_{24}$	0.6	0.7
$S_{25}$	0.7	0.2	$S_{26}$	0.9	0.2
$S_{27}$	0.9	0.6	$S_{28}$	0.7	0.6
$S_{29}$	0.2	0.7	$S_{30}$	1	0.9
$S_{31}$	1	1	$S_{32}$	0.6	1
$S_{33}$	0.9	0.4	$S_{34}$	1	0.4
$S_{35}$	1	0.7	$S_{36}$	0.8	0.2
$S_{37}$	0.5	0.1	$S_{38}$	0.2	0.9
$S_{39}$	0.4	0.4	$S_{40}$	0.5	0.9
$S_{41}$	0.1	0.7	$S_{42}$	0.5	0.4

Note: Two or more wireless sensors will never occupy the same position; each position is allocated only one wireless sensor, as illustrated in Figure 1.

**Table 3 sensors-23-02345-t003:** Centroids after clustering.

Centroids	x-Axis	y-Axis	Centroids	x-Axis	y-Axis
$C_{1}$	0.38	0.24	$C_{2}$	0.3636	0.8
$C_{3}$	0.8769	0.3	$C_{4}$	0.8875	0.75

**Table 4 sensors-23-02345-t004:** Pairwise centroid-to-centroid distance based on Cartesian distances.

	$C_{1}$	$C_{2}$	$C_{3}$	$C_{4}$
$C_{1}$	0	0.5602	**0.5005**	0.7195
$C_{3}$	0.5005	0.7166	0	**0.4501**
$C_{4}$	0.7195	**0.5262**	0.4501	0
$C_{2}$	0.5602	0	0.7166	0.5262

**Table 5 sensors-23-02345-t005:** Joint trajectory schedule for global transmission time period *T* and optimal number of clusters *K*, where the UAV period $t^{}=\left\{\left(mod\left\{T,K\right\}\left|mod\left\{T,K\right\}\neq 0\right.\right)\vee \left(K\left|mod\left\{T,K\right\}=0\right.\right)\right\}$.

**Global period** *T*	1	2	3	4	5 *…*
**UAV period $t^{}$**	1	2	3	4	1 *…*
**Clusters $C_{k}\left|k\in K\right.$**	$C_{1}$	$C_{3}$	$C_{4}$	$C_{2}$	$C_{1}$ *…*
**No. members** $N_{k}=\left|C_{k}\right|$	10	13	8	11	10 *…*
**Members $S_{n}^{\left(i,k\right)}$**	$S_{5}^{\left(1,1\right)}$, $S_{11}^{\left(2,1\right)}$, $S_{13}^{\left(3,1\right)}$, $S_{16}^{\left(4,1\right)}$, $S_{21}^{\left(5,1\right)}$, $S_{22}^{\left(6,1\right)}$, $S_{23}^{\left(7,1\right)}$, $S_{37}^{\left(8,1\right)}$, $S_{39}^{\left(9,1\right)}$, $S_{42}^{\left(10,1\right)}$	$S_{1}^{\left(1,3\right)}$, $S_{6}^{\left(2,3\right)}$, $S_{8}^{\left(3,3\right)}$, $S_{12}^{\left(4,3\right)}$, $S_{14}^{\left(5,3\right)}$, $S_{15}^{\left(6,3\right)}$, $S_{17}^{\left(7,3\right)}$, $S_{18}^{\left(8,3\right)}$, $S_{25}^{\left(9,3\right)}$, $S_{26}^{\left(10,3\right)}$, $S_{33}^{\left(11,3\right)}$, $S_{34}^{\left(12,3\right)}$, $S_{36}^{\left(13,3\right)}$	$S_{3}^{\left(1,4\right)}$, $S_{4}^{\left(2,4\right)}$, $S_{20}^{\left(3,4\right)}$, $S_{27}^{\left(4,4\right)}$, $S_{28}^{\left(5,4\right)}$, $S_{30}^{\left(6,4\right)}$, $S_{31}^{\left(7,4\right)}$, $S_{35}^{\left(8,4\right)}$	$S_{2}^{\left(1,2\right)}$, $S_{7}^{\left(2,2\right)}$, $S_{9}^{\left(3,2\right)}$, $S_{10}^{\left(4,2\right)}$, $S_{19}^{\left(5,2\right)}$, $S_{24}^{\left(6,2\right)}$, $S_{29}^{\left(7,2\right)}$, $S_{32}^{\left(8,2\right)}$, $S_{38}^{\left(9,2\right)}$, $S_{40}^{\left(10,2\right)}$, $S_{41}^{\left(11,2\right)}$	$S_{5}^{\left(1,1\right)}$, $S_{11}^{\left(2,1\right)}$, $S_{13}^{\left(3,1\right)}$, $S_{16}^{\left(4,1\right)}$, $S_{21}^{\left(5,1\right)}$, $S_{22}^{\left(6,1\right)}$, $S_{23}^{\left(7,1\right)}$, $S_{37}^{\left(8,1\right)}$, $S_{39}^{\left(9,1\right)}$, $S_{42}^{\left(10,1\right)}$

**Table 6 sensors-23-02345-t006:** Power allocation factors at wireless sensors for transmitting messages, arranged according to subclusters.

$C_{1}$	$\alpha_{S_{5}^{\left(1,1\right)}}=0.18182$, $\alpha_{S_{11}^{\left(2,1\right)}}=0.16364$, $\alpha_{S_{13}^{\left(3,1\right)}}=0.14545$, $\alpha_{S_{16}^{\left(4,1\right)}}=0.12727$, $\alpha_{S_{21}^{\left(5,1\right)}}=0.10909$, $\alpha_{S_{22}^{\left(6,1\right)}}=0.090909$, $\alpha_{S_{23}^{\left(7,1\right)}}=0.072727$, $\alpha_{S_{37}^{\left(8,1\right)}}=0.054545$, $\alpha_{S_{39}^{\left(9,1\right)}}=0.036364$, $\alpha_{S_{42}^{\left(10,1\right)}}=0.018182$
$C_{2}$	$\alpha_{S_{2}^{\left(1,2\right)}}=0.16667$, $\alpha_{S_{7}^{\left(2,2\right)}}=0.15152$, $\alpha_{S_{9}^{\left(3,2\right)}}=0.13636$, $\alpha_{S_{10}^{\left(4,2\right)}}=0.12121$, $\alpha_{S_{19}^{\left(5,2\right)}}=0.10606$, $\alpha_{S_{24}^{\left(6,2\right)}}=0.090909$, $\alpha_{S_{29}^{\left(7,2\right)}}=0.075758$, $\alpha_{S_{32}^{\left(8,2\right)}}=0.060606$, $\alpha_{S_{38}^{\left(9,2\right)}}=0.045455$, $\alpha_{S_{40}^{\left(10,2\right)}}=0.030303$, $\alpha_{S_{41}^{\left(11,2\right)}}=0.015152$
$C_{3}$	$\alpha_{S_{1}^{\left(1,3\right)}}=0.14286$, $\alpha_{S_{6}^{\left(2,3\right)}}=0.13187$, $\alpha_{S_{8}^{\left(3,3\right)}}=0.12088$, $\alpha_{S_{12}^{\left(4,3\right)}}=0.10989$, $\alpha_{S_{14}^{\left(5,3\right)}}=0.098901$, $\alpha_{S_{15}^{\left(6,3\right)}}=0.087912$, $\alpha_{S_{17}^{\left(7,3\right)}}=0.076923$, $\alpha_{S_{18}^{\left(8,3\right)}}=0.065934$, $\alpha_{S_{25}^{\left(9,3\right)}}=0.054945$, $\alpha_{S_{26}^{\left(10,3\right)}}=0.043956$, $\alpha_{S_{33}^{\left(11,3\right)}}=0.032967$, $\alpha_{S_{34}^{\left(12,3\right)}}=0.021978$, $\alpha_{S_{36}^{\left(13,3\right)}}=0.010989$
$C_{4}$	$\alpha_{S_{3}^{\left(1,4\right)}}=0.22222$, $\alpha_{S_{4}^{\left(2,4\right)}}=0.19444$, $\alpha_{S_{20}^{\left(3,4\right)}}=0.16667$, $\alpha_{S_{27}^{\left(4,4\right)}}=0.13889$, $\alpha_{S_{28}^{\left(5,4\right)}}=0.11111$, $\alpha_{S_{30}^{\left(6,4\right)}}=0.083333$, $\alpha_{S_{31}^{\left(7,4\right)}}=0.055556$, $\alpha_{S_{35}^{\left(8,4\right)}}=0.027778$

## Data Availability

The data used in this study were randomly generated.

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
