# Peer review of "Sensor Clustering Using a K-Means Algorithm in Combination with Optimized Unmanned Aerial Vehicle Trajectory in Wireless Sensor Networks"

_sensors, 2023, doi:10.3390/s23042345_

Round 1

Reviewer 1 Report

The paper presents a sensor Clustering with K-means for Joint UAV Trajectory

Enables Wireless Sensor Networks. While the paper is technically sound, there are many issues and unclear points present in the paper. The authors may consider the following comments to improve the quality of the paper.

1.  Paper needs some proofreading. There are a multiple grammatical and typing errors. Please review the paper.

2.  The author uses the gap statistic method to get the optimal number of clusters, and directly gives the optimal number of clusters as four through Figure 2, without giving the reason, please explain.

3.  In the paper, the authors use the k-means clustering algorithm. There are a variety of clustering algorithms, and there exist many clustering algorithms that are better than k-means, Why did the authors choose algorithm?

4.  The author has repeatedly used the pictures in the references to illustrate the problem in the paper, and it is recommended that the author draw pictures in this paper if necessary.

5.  Discussion of the results requires a more in-depth analysis. The authors only report the results and the analysis goes not into depth. Therefore, this weakens the contribution of the paper.

Author Response

Manuscript ID: sensors-2171476    

Article Title: “Sensor Clustering with K-means for Joint UAV Trajectory Enables Wireless Sensor Networks”

Dear Editor,

We are grateful to the Editors and the Reviewers who gave us a chance to improve our work. We express our deep sense of gratitude to all the reviewers for their valid comments and constructive suggestions, which have helped us to improve the quality and presentation of the manuscript. Appropriate changes have been made in the major revised manuscript to improve the manuscript. Please see the \hl{highlighted text} in the manuscript.

We are uploading (a) our point-by-point response to the comments (below) (response to reviewers), (b) an updated manuscript with yellow highlighting indicating changes (Supplementary Material for Review).

Best regards,

Thanh-Long Nguyen et al.

Corresponding author: [email protected]

Reviewer 2 Report

The paper contains a valuable contribution. The subject is within the scope of the journal and the objective of research is well stated. However, some clarifications about the underlying hypothesis/scope/findings are needed.

In the opinion of this Reviewer the manuscript deserves to be published once the Author takes into account the raised issues.

Introduction / Literature review

1.       The research scope is clear as well as the literature review. Anyway, the authors should better highlight the innovative aspects of their work in the manuscript.

What are the advantages / findings in the proposed paper, which are not covered by other studies/reviews? My suggestion is that the authors dedicate a paragraph intitled “Related work” instead making a continuous comparison along the paper.

WSN model

2.       The authors refer to a figure in another paper. This is confusing. However I can’t see difference between the two distribution. Both of them are random and in the paper [18] no number are given.

3.       The authors claim that they use a 3D positioning, but they impose the z-coordinate equal to 0. Please explain this point.

WSN clustering

4.       Row 118: why UAV cannot fly close to each sensor to collect data? Please clarify this point.

Minor

5.       Check the bit rate unity of measurement.

6.       Extensive editing of English language and style required. The paper should be carefully rechecked.

Author Response

Manuscript ID: sensors-2171476    

Article Title: “Sensor Clustering with K-means for Joint UAV Trajectory Enables Wireless Sensor Networks”

Dear Editor,

We are grateful to the Editors and the Reviewers who gave us a chance to improve our work. We express our deep sense of gratitude to all the reviewers for their valid comments and constructive suggestions, which have helped us to improve the quality and presentation of the manuscript. Appropriate changes have been made in the major revised manuscript to improve the manuscript. Please see the highlighted text in the manuscript.

We are uploading (a) our point-by-point response to the comments (below) (response to reviewers), (b) an updated manuscript with yellow highlighting indicating changes (Supplementary Material for Review).

Best regards,

Thanh-Long Nguyen et al.

Corresponding author: [email protected]

Reviewer 3 Report

The manuscript presents a clear structure and the presentation of simulated results seems to be suitable.

However, I have the following observations.

If possible, you should avoid using abbreviations in the title. Define abbreviations and acronyms the first time they are used in the text, even after they have been defined in the abstract.

The abstract should be concise and inform the readers of background, research question, hypothesis, methodology, the main results and conclusions of research presented. The main results obtained must be highlighted in the abstract.

Author Response

(The authors gave the same response as above.)

Round 2

Reviewer 1 Report

All the questions raised by the reviewer have been addressed, and I recommend the acceptance of this paper.

Reviewer 2 Report

Authors have properly enriched their work, by addressing each comment in a suitable way. The paper turns out to be notably improved.